# Living with the pathogenic chytrid fungus: Exploring mechanisms of coexistence in the harlequin toad *Atelopus cruciger*

Onil Ballestas[1], Margarita Lampo[1]*, Diego Rodríguez[2]

**1** Centro de Ecología, Instituto Venezolano de Investigaciones Científicas, Caracas, Venezuela,
**2** Laboratorio de Ecología y Evolución, Instituto de Zoología y Ecología Tropical, Universidad Central de Venezuela, Caracas, Venezuela

* mlampo@gmail.com

## Abstract

Chytridiomycosis, a disease caused by the fungus *Batrachochytrium dendrobatidis* (*Bd*), has been linked with the disappearance of amphibian populations worldwide. Harlequin toads (*Atelopus*) are among the most severely impacted genera. Two species are already considered extinct and most of the others are at high risk of extinction. The recent rediscovery of harlequin toad populations coexisting with *Bd* suggest that the pathogen can maintain enzootic cycles at some locations. The mechanisms promoting coexistence, however, are not well understood. We explore the dynamics of *Bd* infection in harlequin toads by modeling a two-stage host population with transmission through environmental reservoirs. Simulations showed that variations in the recruitment of adults and the persistence of zoospores in the environment were more likely to drive shifts between extinction and coexistence than changes in the vulnerability of toads to infection with *Bd*. These findings highlight the need to identify mechanisms for assuring adult recruitment or minimizing transmission from potential reservoirs, biotic or abiotic, in recovering populations.

## Introduction

Chytridiomycosis, a disease caused by the chytrid fungi, *Batrachochytrium dendrobatidis* (*Bd*) and *B. salamandrovirans*, has been linked to the local extirpation and extinction of amphibian populations worldwide, although the exact number of species affected remains controversial [1, 2]. *Bd* disrupts normal skin function and ion homeostasis causing death in many infected anurans [3–7]. Despite its widespread distribution in amphibian populations, *Bd* remains a serious threat to global amphibian biodiversity. New epidemic events can be triggered by novel climatic scenarios and drive remnant population to extinction [8]. On the other hand, the global amphibian trade continues to translocate chytrid lineages with panzootic potential [9].

Harlequin toads (*Atelopus*) have been one of the most affected taxa by chytridiomycosis, with 82 of 98 described species in a threatened category (VU, EN, CR), and two considered extinct (EX) [10]. Their limited geographic ranges [11] make most harlequin toad species

**Data Availability Statement:** All relevant data are within the paper and its Supporting information files.

**Funding:** The funders had no role in study design, data collection and analysis, decision to publish, or preparation of the manuscript.

**Competing interests:** The authors have declared that no competing interests exist.

highly susceptible to extinction, particularly in highlands, from where more than 75% of the species disappeared [12]. Some species of harlequin toads have been rediscovered in recent years (*A. varius* [13], *A. nepiozomus* [14], *A. palmatus* [14], *A. bomolochos* [15], *A. ignescens* [16], *A. nanay* [17], *A. longirostris* [18] and *A. carrikeri* [19]), and few populations appear to be recovering despite the presence of *Bd* in some of its individuals [20–23]. The mechanisms that allow these populations to coexist with the fungus are not fully understood. A decrease in *Bd* pathogenicity or transmission, or an increase in host resistance or recruitment can potentially promote coexistence, but the parameters describing these processes are unknown for most recovering populations. Recent studies in *A. varius* suggest that epizootic-enzootic transitions in recovering populations from Panama are not likely to be associated with *Bd* attenuation [23].

*Atelopus cruciger* is one of the few species of harlequin toads currently known to coexist with the *Bd*. This species was widespread and abundant in pristine streams across more than 4,000 km$^2$ in northern Venezuela but disappeared from most of its habitats during the early 80s [24]. The presence of fungus in the last individuals collected in 1986 suggested chytridiomycosis as the most likely cause of its disappearance [25]. In 2003, populations were detected at two nearby lowland locations [26]. Pathogen-induced mortality, infection rates or population recruitment rates, estimated from capture histories collected for seven years at one locality, indicated that harlequin toads have coexisted in endemic equilibrium with *Bd* for more than a decade, despite evidence suggesting that this fungus continues to be highly lethal for *A. cruciger* [20, 21]. The authors hypothesized that because transmission rates are low, the rapid recruitment of healthy adults offsets the population losses from the lethal chytridiomycosis. However, underpinning the mechanisms that determine the infection outcome at a population-level requires context-specific modeling.

Mathematical modeling of amphibian infections with *Bd* have demonstrated that the fungus' ability to survive in abiotic or biotic reservoirs is a critical factor driving host populations to extinction [27–30]. Although one study suggests that zoospores persist year-round in the environment [31], the dynamics of free-living *Bd* is largely unknown. Other models, on the other hand, suggest that host tolerance and resistance to chytridiomycosis are the most important mechanisms affecting the of risk extinction of infected populations [29]. Hypotheses that explain the recovery of some frog populations in lowlands invoke an attenuated lethality of the pathogen in warm habitats [32, 33]. To explore possible mechanisms promoting the stable persistence of *A. cruciger* populations with *Bd*, we developed a mathematical model for a two life stage host population with transmission through environmental reservoirs following Louca et al. [28]. Using parameters values estimated from wild populations of *A. cruciger* [21] and experimental infections in *A. zeteki* [34], we identified mechanisms and parameter thresholds leading to transitions between possible outcomes —coexistence, host extinction or pathogen extinction— by numerical simulations of different scenarios. We investigated two hypotheses, one invoking diminished *Bd* virulence or host vulnerability and the other low transmission due to low zoospore survival in environmental reservoirs. The identification of mechanisms of population recovery can shed light on the design of mitigation actions for this and other critically endangered harlequin frog species.

## Materials and methods

### Infection model

We used demographic and epidemiological parameters estimated from sighting histories of post-metamorphic individuals in a wild population of *A. cruciger* [21, 35]. Parameters

describing the *Bd* intra-host dynamics, which were not available for *A. cruciger*, were approximated using estimates from other species or from *in vitro* cultures (Table 1 in S1 Appendix).

Using difference equations, we modeled infection in a population structured in two life stages, pre- and post-metamorphic frogs (from here referred as tadpoles, *T*, and adults, *A*) following Louca et al. [28]. We used weekly time intervals, *t*, to simulate the *Bd*-frog dynamics for 2,000 weeks, which is approximately equivalent to 38 years or 10 generations. We assumed that tadpoles mature in a maximum of 12 weeks and post-metamorphic frogs live a maximum of 78 weeks. We kept track of the age, *j*, and the number of zoospores, *k*, in each individual and denoted by $P(t) = \sum_{j=1}^{12} p(j, t)$ the total numbers of tadpoles and by $A(t) = \sum_{j=1}^{78} a(j, t)$ the total number of adults, respectively, at time *t*, where $p(j, t)$ and $a(j, t)$ are the number of tadpoles and adults, respectively, in age *j* at time *t*.

The number of zoospores harbored by a tadpole and an adult individual of age *j*, at time *t*, were denoted by $z_P(j, t)$ and $z_A(j, t)$, respectively. Therefore, $Z_A(t) = \sum_{j=1}^{78} a(j, t) z_A(j, t)$ and $Z_P(t) = \sum_{j=1}^{12} p(j, t) z_P(j, t)$ correspond to the total number of zoospores in the adult and tadpole populations, respectively, at time *t*. The degree of infection of tadpoles was estimated as the mean number of zoospores per individual at time *t*, $I_P(t) = Z_P(t)/P(t)$ or $I_A(t) = Z_A(t)/A(t)$, and the number of zoospores in the abiotic reservoir was denoted, $Z_E(t)$.

The simulations included the following events: reproduction of adults, maturation of tadpoles, survival, acquisition of zoospores from the reservoir and zoospore release into the reservoir, and zoospore intra-host dynamics (Fig 1). The temporal dynamics in the number of

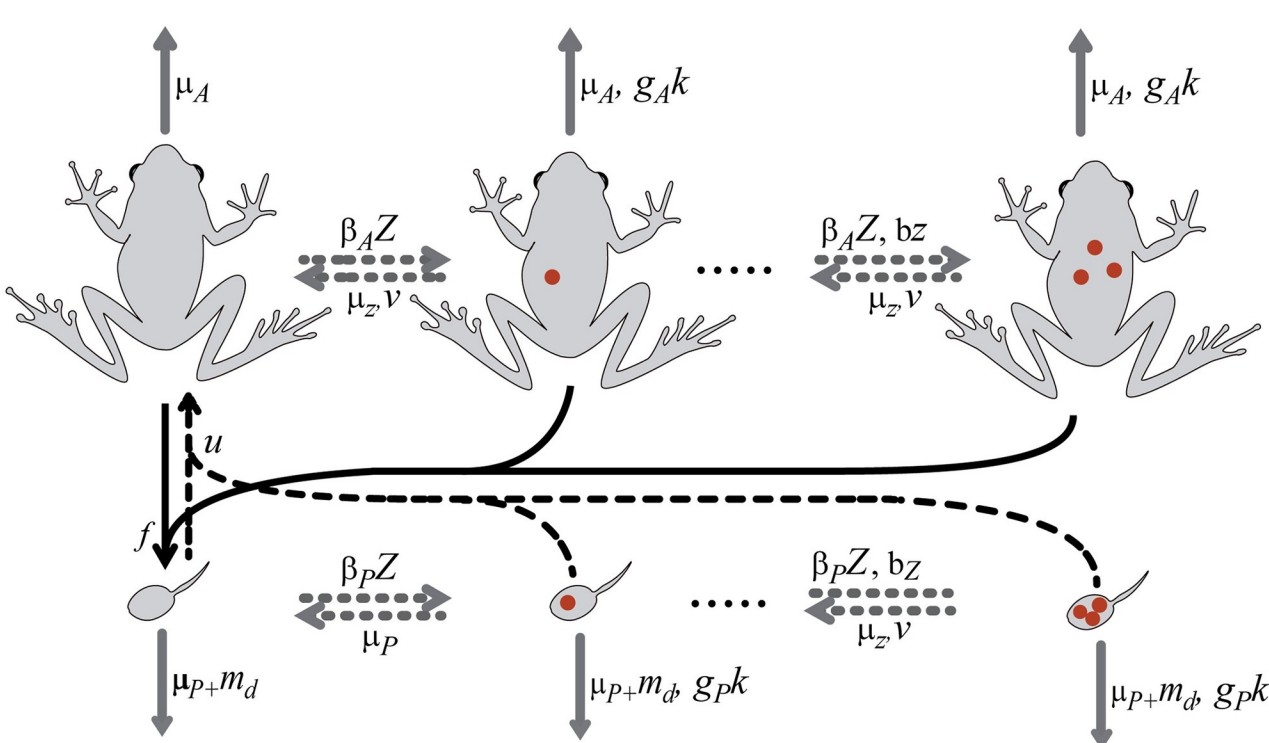

**Fig 1. Schematic of the dynamics of *Bd* infection in reproductive adults and tadpoles.** Solid black arrows correspond to reproduction, dashed black arrows to recruitment of tadpoles into reproductive adults, grey dashed arrows to new infections or within-host disease dynamics, grey solid arrows indicate disease-independent or disease-induced deaths.

tadpoles and adults are described by,

$$P(t+1) = \sum_{j=1}^{12} p(j,t)(1 - m_P(P(t), z_P(j,t))) + f(t)\frac{A(t)}{2} - \sum_{j=1}^{12} u(j)p(j,t) \tag{1}$$

$$A(t+1) = \sum_{j=1}^{78} a(j,t)(1 - m_A(z_A(j,t))) + \sum_{j=1}^{12} u(j)p(j,t) \tag{2}$$

Function $f(t)$ describes the population reproduction as a unimodal function with a maximum value during the dry season and a spread proportional to the parameter $S$ (eqn. A1 in S1 Appendix). The function $u(j)$ defines maturation as an exponentially increasing function of age (eqn. A2 in S1 Appendix). The tadpole mortality, $m_P(P,k)$, increases with their density and the number of zoospores harbored (eqn. A3 in S1 Appendix), whereas adult mortality, $m_A(k)$, increases linearly only with the number of zoospores harbored (eqn. A4 in S1 Appendix).

The intra-host dynamics is described by the changes in the degree of infection in tadpoles and adults as follows,

$$I_P(t+1) = I_P(t) + \beta_P \frac{Z_E(t)}{P(t) + A(t)} + I_P(t)(b_z - \mu_Z)(1 - \vartheta) \tag{3}$$

$$I_A(t+1) = I_A(t) + \beta_A \frac{Z_E(t)}{P(t) + A(t)} + I_A(t)(b_z - \mu_Z)(1 - \vartheta) \tag{4}$$

where $\beta_P$ and $\beta_A$ are the rates of zoospore acquisition of tadpoles or adults from the reservoir, $\mu_Z$ is the intra-host per capita death rate of zoospores, $b_Z$ is the intra-host per capita birth rate of zoospores and $v$ the rate of release of zoospores into the abiotic reservoir. The temporal dynamics of zoospores in the abiotic reservoir are defined by,

$$Z_E(t+1) = Z_E(t)(1 - \mu_E) + \vartheta(Z_P(t) + Z_A(t)) - \beta_P \frac{Z_E(t)}{P(t) + A(t)}P(t) - \beta_A \frac{Z_E(t)}{P(t) + A(t)}A(t) \tag{5}$$

where $\mu_E$ is the rate of mortality of zoospores in the abiotic reservoir.

## Simulations

We modeled two scenarios: i) only adults can be infected by *Bd* and ii) both tadpoles and adults are susceptible to infection. The differences between the two simulated scenarios allowed us to explore the potential effect of the infection in tadpoles, given that tadpoles have not been observed in the field for many harlequin frog species. Initial values of the number of tadpoles $p(j,t)$ and adults $a(j,t)$ and the number of zoospores harbored in tadpoles $z_P(j,t)$ and adults $z_A(j,t)$ were chosen intuitively. Simulations were run for 2,000 weeks (approximately 10 generations), when the trajectories tended to stabilize. Long-term dynamics of infection did not vary with small variations on these initial values.

## Extinction risks

By simultaneously varying pairs of parameters while fixing all others, we explored the parameter space to identify the thresholds that define the transitions between the three possible outcomes (host extinction, coexistence, and pathogen extinction). For each tuple, we ran 8,281 simulations. We chose parameters affecting the postulated mechanisms in the hypotheses; that

is to say, the death induced mortality ($g_A$) (attenuated virulence hypothesis), and the acquisition rate of zoospores from the reservoir ($\beta_A$) and life expectancy of the free zoospore (low transmission hypothesis). We also explored variations in the survival of tadpoles, the intensity of the density-dependent mortality in tadpoles and the seasonality of the recruitment of individuals to assess the potential effect of population compensatory mechanisms in the possible outcomes.

For each tuple, the outcome was defined according to the following criteria: i) coexistence, if $\sum_{t=1,990}^{2,000} A(t) > 0$ and $\sum_{t=1,990}^{2,000} I_A(t) > 0$ or $\sum_{t=1,990}^{2,000} I_P(t) > 0$; ii) extinction, if $\sum_{t=1,990}^{2,000} A(t) = 0$; and iii) parasite extinction, if $\sum_{t=1,990}^{2,000} A(t) > 0$, $\sum_{t=1,990}^{2,000} I_P(t) = 0$ and $\sum_{t=1,990}^{2,000} I_A(t) = 0$. The results for each tuple are presented as a phase diagram.

## Results

The number of tadpoles and adults showed seasonal stable oscillations for 2,000 weeks (Fig 2). The number of adults varied between 10 and 320 (Fig 2A), the number of tadpoles between 0 and 12,000 (Fig 2B) and the mean zoospore burden between 0 and 50,000 (Fig 2C). Tadpole abundance showed a bimodal distribution with peaks in the beginning and the middle of the year (Fig 3A). The first peak results from an increase in the fecundity, shortly after the dry season begins in December, and the second peak occurs after a drastic decline in the zoospore burden of adults during the dry season. Adult abundance showed a single peak between September and October, when most tadpoles had metamorphosed (Fig 3A). Zoospore burden in adults closely tracked the number of free zoospores in the environment. The highest numbers of zoospores in the environment and within the host occurs at the beginning of the year, when tadpole abundance is high (Fig 3).

Coexistence (blue) or failed *Bd*-establishment (green) were possible outcomes in all parameter spaces explored (Fig 4A–4J). Extinction (red), however, appeared as a probable outcome only if survival of pre-adults was low (Fig 4G and 4H) or fecundity became highly seasonal (Fig 4I and 4J). If tadpoles were not involved in transmission (left panel), *Bd* could not establish in most scenarios, unless zoospore persisted in the environment for more than 0.6 weeks (Fig 4A, 4C and 4E; left panel). If, on the other hand, tadpoles got infected and were a source of infection for reproductive individuals, *Bd* establishment was possible, even if the free zoospore persisted in the environment for less than a day (Fig 4B, 4D and 4F). As fecundity became more seasonal, *Bd* establishment turned more difficult. *Bd* failed to establish if its persistence in the environment was too low. If, on the contrary, zoospores survived for too

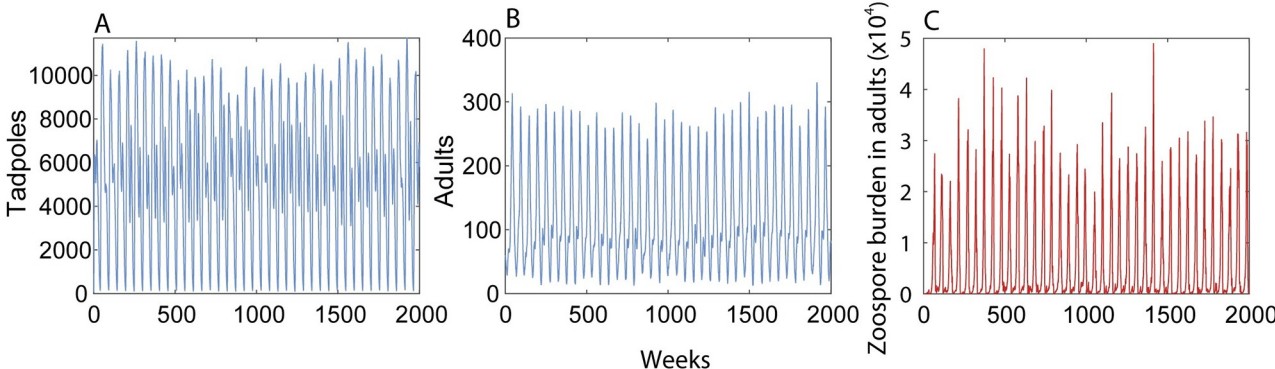

**Fig 2. Long-term dynamics of the *Atelopus cruciger–Bd* interaction.** Tadpoles and adults can get infected but only adults develop lethal chytridiomycosis. Simulations for 2,000 weeks show stable oscillations in the abundance of tadpoles, adults, and in the zoospore burden.

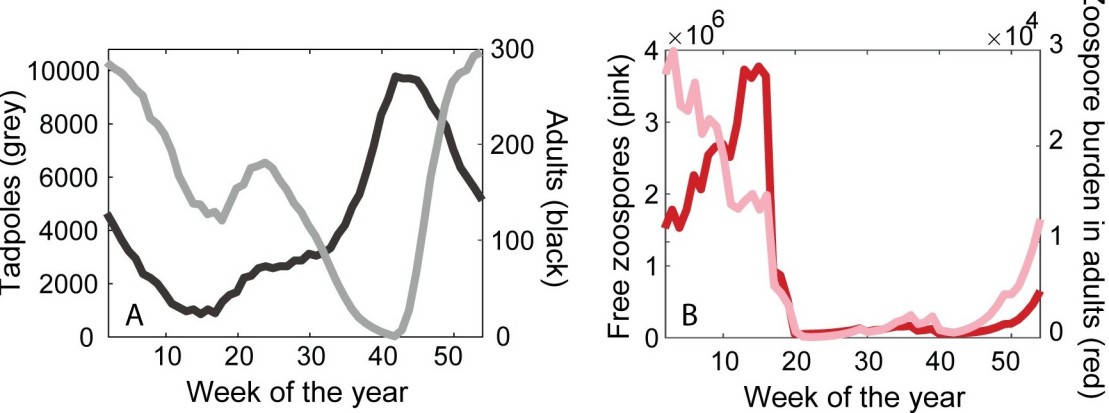

**Fig 3. Yearly dynamics of the *A. cruciger–Bd* interaction.** Tadpoles and adults can get infected but only adults can develop lethal chytridiomycosis. (A) The number of tadpoles and adults. (B) Number of free zoospores in the environment and the *Bd* burden in adults.

long outside their hosts (*e.g.* > 1.4 weeks) extinction was the most probable outcome (Fig 4I and 4J).

Hosts are more vulnerable to infection, if the lethal burden is lowered (Fig 5A) or if the reproductive rate of the pathogen is increased (Fig 5B). In both cases, *Bd*-induced mortality will increase or the life expectancy of the host will be decreased. Changes in the vulnerability of post-metamorphic toads within the explored range did not drive host populations to extinction, but could prevent *Bd* from establishing. If tadpoles were involved in transmission, a decrease in the lethal zoospore burden or an increment in the reproductive rate of the pathogen showed no effect on the probability of coexistence (Fig 4B). If tadpoles did not get infected, *Bd* establishment was unlikely for values of *Bd*-induced mortality close to $10^{-6}$/zoospore-week and zoospore life expectancy below 0.6 weeks (Fig 4A). A similar effect was observed with the transmission rates from the reservoir. Variations within the explored range in the proportion of zoospores acquired weekly from the environment did not result in extinction (Fig 4C and 4D), but *Bd* establishment was not possible if tadpoles were not involved in transmission and the zoospore could not persist in the environment for more than 3–4 days (Fig 4C).

Extinction was the most likely outcome, however, if the weekly survival of the pre-adults was below 60—70% (Fig 4G and 4H) or if adult reproduction was highly seasonal (*e.g.* restricted to less than 23 weeks each year) (Fig 4I and 4J). A long interruption of recruitment of healthy adults could drive the host population to extinction, even if tadpoles are not exposed to infection. This suggests that recruitment of healthy adults is the mechanism most likely to prevent the extinction of the frog population, even if tadpoles are infected with *Bd*. In highly seasonal environments coexistence is only possible if zoospores can persist for 1–2 weeks in the environment (Fig 4I and 4J). Establishment of *Bd* depends on the capacity of zoospores to subsist outside its host, but a prolonged persistence of zoospores in the environment can drive the toad population to extinction.

## Discussion

The high impact of chytridiomycosis at high elevations [1, 36–38], and the recent discovery of recovering populations in lowlands [26, 39, 40] suggest that warm temperatures may promote the coexistence of some amphibian species with the pathogenic fungus *Bd* [41, 42]. The mechanisms at play, however, can be diverse and difficult to pinpoint. Simulation results showed that

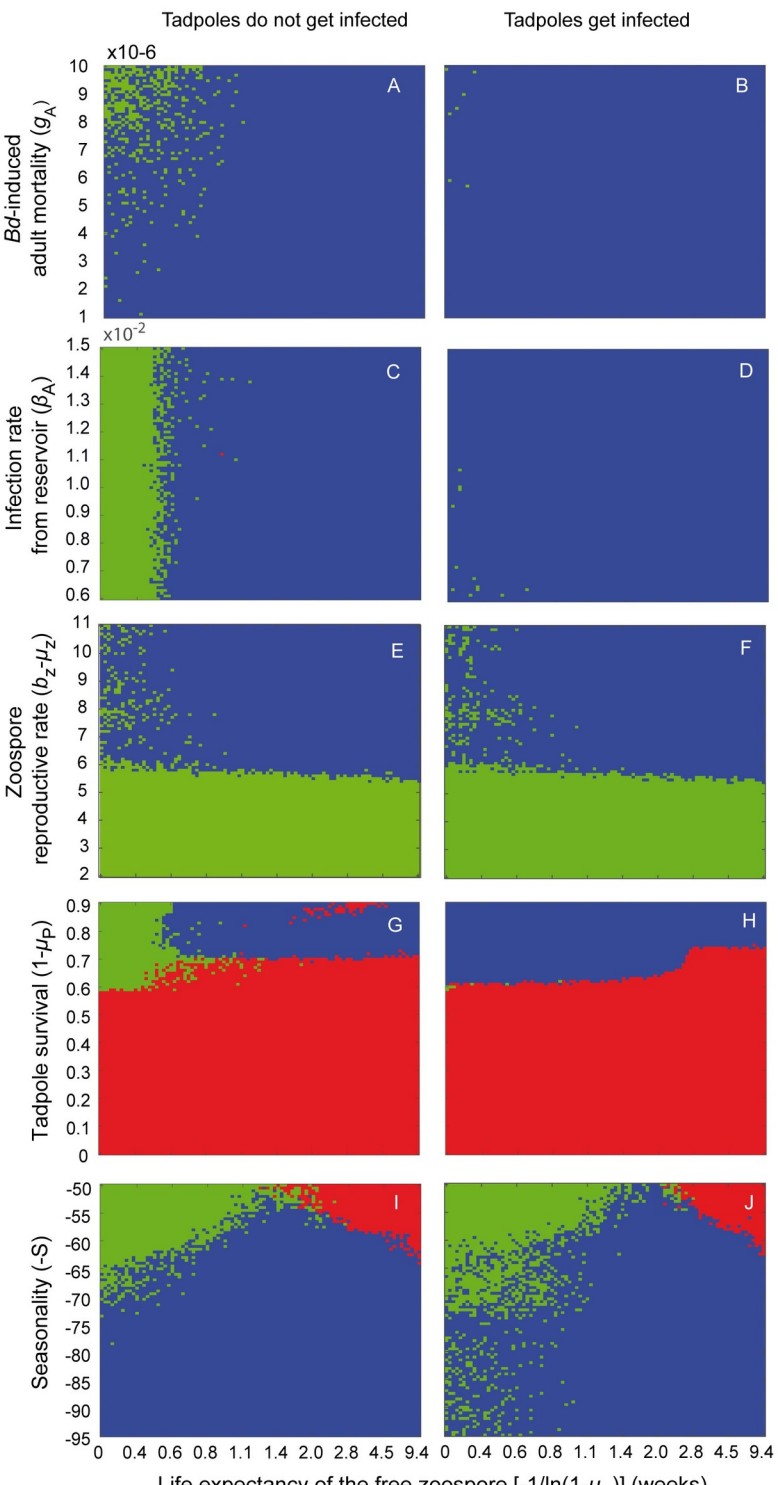

**Fig 4. Infection outcomes expected for pairs of parameters.** Each pixel represents a simulation of 2,000 weeks. Blue pixels represent combinations of parameters for which coexistence is the most likely outcome, green denotes pairs for which *Bd* is most likely to fail in establishing and red denotes pairs for which *Bd* is more likely to extirpate the toad population. In the left panel (A,C,E,G,I), tadpoles do not get infected. In the right panel (B,D,F,H,J), tadpoles get infected but do not develop lethal chytridiomycosis. Population parameters fixed are described in Table 1 in S1 Appendix.

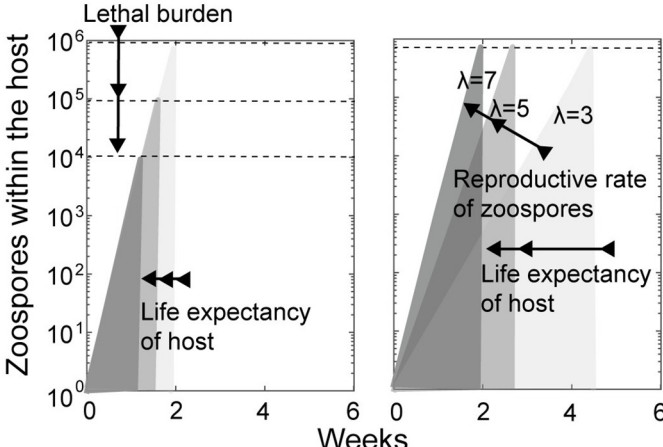

**Fig 5. Relationship between zoospore lethal burden, reproductive rate and the life expectancy of host (λ).** The dashed line identifies the zoospore concentration for which host survival is zero. The vertical projections of the intercept of the number of zoospores with the lethal zoospore load indicates the life expectancy of the host for each value of $\lambda$.

parameters determining adult recruitment and the life expectancy of the free zoospores are more likely to drive populations of harlequin toads from endemic coexistence to extinction, than those affecting the vulnerability of adults to chytridiomycosis. Extinction was a possible outcome of infection, only in scenarios in which pre-adult survival was low, recruitment of post-metamorphic individuals was highly seasonal, or the zoospore could persist for more than a week in the environment. Moderate changes in the life expectancy of infected toads, on the contrary, had little effect on the probability of extinction.

Chytridiomycosis is highly lethal for harlequin toads. In captivity, adults of *A. zeteki* and *A. varius* die within 4–5 weeks after experimental inoculation with *Bd* [23, 34]. In wild populations, the life expectancy of infected adults of *A. cruciger* is estimated to be 2–3 weeks [21]. It is not known whether infected toads can clear infection. Experimentally inoculated adults of *A. zeteki* and *A. varius* showed 100% mortality [23, 34]. On the other hand, naturally infected toads from wild populations are rarely recaptured. Although two *A. cruciger* adults appeared to have cleared infection during a seven year study [21], environmental contamination could not be ruled out, given the low genome equivalents detected (<10 genome equivalents). If *Bd* attenuation or an increase in toad susceptibility can only increment the life expectancy of infected toads, but not their chances of fully recovering from infection, as we have modeled, their impact on the extinction probability of populations is negligible. Increasing the lethal burden two orders of magnitude or reducing by half the reproductive rate of zoospores within toads can only increase the life expectancy of an infected toad in few weeks (Fig 5). In contrast, harlequin toads tend to breed yearly. In captivity, where feeding rates are high, oocyte maturation can take several months. Because the time scale on which infection, disease development and death occur is much faster than their breeding cycle, an increase in the life expectancy of infected toads will have little impact on the reproductive output of the population. This means that changes in the susceptibility of harlequin toads to chytridiomycosis or in the pathogenicity of *Bd* would have to increase substantially the probability of fully recovering from infection to have an effect on the probability of extinction of populations coexisting with *Bd*. Similarities in the pathogenicity and infectivity of *Bd* isolates recovered during epizootic events and from recovering populations ruled out pathogen attenuation as the mechanism responsible for the

transition between epizootic and enzootic dynamics in amphibian populations from Panama [23].

Harlequin toads tend to be short-lived in the wild. Most adults live long enough to reproduce only during one breeding season [20, 21, 43]. Consequently, the reproductive population must be replaced every year for their abundance to remain stable. Populations with rapid turnover rates depend on a sustained recruitment of individuals to persist, a mechanism that is warranted by the high fecundity of harlequin toads. In the presence of a highly pathogenic *Bd*, infected harlequin toads are not likely to reproduce. Amplexus can last months but infected toads die few weeks after infection. In the model, we assumed that exposure to *Bd* occurs when individuals are adults. Thus, recruitment of healthy juveniles is the main mechanism maintaining the reproductive output of the population. Although juveniles can get infected, the age-prevalence relationship in wild caught toads suggests that the prevalence of infection is much lower in juveniles [21]. Thus, a large recruitment of healthy juveniles can play a major role in securing the reproductive output of the population. The post-epidemic recovery of infected populations of some other amphibian species has been linked to a high recruitment of healthy adults [44, 45]. The high sensitivity of the transitions between coexistence and extinction to parameters affecting recruitment in simulated infections suggests this mechanism as the most likely associated with the capacity of harlequin toad populations to establish an enzootic equilibrium with the chytrid fungus.

While genetic evidence increasingly supports the global emergence of chytridiomycosis as a novel epidemiological event (novel pathogen hypothesis), the spatio-temporal pattern of amphibian declines in many regions more closely relates with climatic events (*i.e.*, [reviewed in [46]]). This means that the ability of populations to withstand an initial outbreak was probably determined by environmental factors. If the probability of persisting with *Bd* depends on variations in the population capacity to recruit rather than their species capacity to resist chytridiomycosis, then we would expect a larger impact of the disease in locations or years when recruitment has been substantially reduced due to climatic events. Highly synchronized disappearance of distant harlequin toad populations in the Venezuelan Andes suggests that extreme droughts may have played a role in the local extinction after the *Bd* initial invasion [47].

Pathogen transmission tends to be low in highly vulnerable hosts because they die shortly after infected. The devastating effect of *Bd* on amphibian populations has been attributed to its ability to survive in alternative hosts, resistant tadpoles or in the environment [27–30]. Model simulations showed that *Bd* establishment is unlikely in highly vulnerable harlequin toads, unless zoospores are able to survive outside adult toads for several days. Moreover, if recruitment is highly seasonal *Bd* establishment becomes more difficult. This indicates that for *Bd* to have successfully established in harlequin toad populations, zoospore persistence outside post-metamorphic toads must be high. The extent to which tadpoles take part in the transmission cycle in harlequin toads is unknown as they are rarely observed in the wild. Tadpoles could increase zoospore persistence if their prevalence of infection is high or, conversely, promote the recruitment of healthy adults, if prevalence is low [28]. Few observations in mountain species suggest that tadpoles attach to rocks in streams with high currents. Mechanisms of *Bd* transmission under these conditions are not clear. If tadpoles do not substantially contribute *Bd* to transmission, then abiotic reservoirs or alternative host become the only mechanisms available for sustaining the *Bd* population over time. Persistence of zoospores in the environment over too long periods can, on the other hand, drive the toad populations to extinction.

The impact of chytridiomicosis on amphibian communities has proved to be context-dependent. Mechanisms promoting coexistence can vary between species, or even populations, depending on pathogen- or host-inherent characteristics or the environment to which they are exposed (reviewed in [46]). A similar modeling study of the infection dynamics of the

Mountain Yellow-legged Frog complex (*Rana muscosa*) indicated, in contrast with our findings, that extinction risk of Mountain Yellow-legged Frog populations is more sensitive to host resistance and tolerance than to the transmission dynamics of *Bd* [29]. Such apparent discrepancies result from different model assumptions derived from contrasting life-histories of Mountain Yellow-legged Frogs and harlequin toads. The Mountain Yellow-legged Frog model assumes that some infected frogs clear infection to become susceptible. That is, not only does the recruitment of healthy juveniles contribute to the susceptible population, but also the presence of infected adults that have cleared infection. This mechanism was not included in our model, in light of the existing evidence suggesting that all infected harlequin toads succumb to chytridiomycosis [21, 23, 34]. Thus, survival of adult harlequin toads do not depend on their ability to tolerate infection but on their chances of escaping infection. On the other hand, Mountain Yellow-legged Frogs appear to have much longer life expectancies and slower population turnover rates than harlequin toads. Tadpoles take several years to mature and can potentially act as a long-lived reservoir for *Bd* infection and adults may survive for many years [29, 30]. Thus, *Bd* establishment in Mountain Yellow-legged Frog populations does not depend on the presence of abiotic reservoirs where zoospores can persist for several weeks [29], as it appears to be the case for harlequin toad populations. These differences highlight the need for context modeling for a better assessment of effective strategies to mitigate chytridiomycosis in amphibian populations.

## Conclusion

Mathematical models can provide insights into mechanisms involved in the persistence of enzootic chytridiomycosis in amphibian populations. Simulations suggest that for harlequin toads, a genus highly susceptible to chytridiomycosis, population transitions between coexistence and extinction are more sensitive to variations in the recruitment of post-metamorphic toads and the persistence of the zoospore in the environments than to changes in the pathogen-induced mortality. Identifying the environmental conditions or mechanisms that reduce the persistence of zoospores outside the host or increase the capacity of population to compensate mortality via recruitment of adults could be important steps toward the effective management of harlequin toad populations that are recovering in the presence of *Bd*.

## Supporting information

**S1 Appendix. Description of the model parameters and functions.** This appendix describes the parameters and functions used for modeling the infection dynamics of the chytrid fungus *B. dendrobatidis* in a population of the harlequin toad *Atelopus cruciger*.
(ZIP)

**S1 File. Simulating the infection dynamics.** A MATLAB script for simulating the infection dynamics of the chytrid fungus *B. dendrobatidis* in a frog population structured in two age-classes.
(TXT)

## Acknowledgments

We are grateful to Celsi Señaris and Hendrik Sulbarán for their comments in early versions of the manuscript and to Santiago Nestares Lampo for optimizing the scripts for parallel computing. The final version of this manuscript benefited from suggestions by an anonymous reviewer.

## Author Contributions

**Conceptualization:** Margarita Lampo.

**Formal analysis:** Onil Ballestas.

**Investigation:** Onil Ballestas.

**Methodology:** Onil Ballestas, Margarita Lampo, Diego Rodríguez.

**Software:** Diego Rodríguez.

**Supervision:** Margarita Lampo.

**Validation:** Onil Ballestas, Margarita Lampo.

**Writing – original draft:** Onil Ballestas.

**Writing – review & editing:** Onil Ballestas, Margarita Lampo, Diego Rodríguez.

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
