## [Decision Letter · Decision Letter 0]

13 Apr 2021

PONE-D-21-04928

Living with the pathogenic chytrid fungus: exploring mechanisms of coexistence in the harlequin toad Atelopus cruciger.

PLOS ONE

Dear Dr. Lampo,

Thank you for submitting your manuscript to PLOS ONE. After careful consideration, we feel that it has merit but does not fully meet PLOS ONE’s publication criteria as it currently stands. Therefore, we invite you to submit a revised version of the manuscript that addresses the points raised during the review process.

There was one referee who was positive about the paper, as I am. It makes a valuable contribution. Ciritiques by the referee include that other studies in the field are not thoroughly considered (introduction, discussion). Including them will certainly improve the value of this contribution. Suggestions for references to be considered are made by the referee.

We look forward to receiving your revised manuscript.

Kind regards,

Stefan Lötters

Academic Editor

PLOS ONE

Journal Requirements:

Additional Editor Comments (if provided):

Reviewers' comments:

Reviewer's Responses to Questions

**Comments to the Author**

1. Is the manuscript technically sound, and do the data support the conclusions?

Reviewer #1: Yes

2. Has the statistical analysis been performed appropriately and rigorously? 

Reviewer #1: I Don't Know

3. Have the authors made all data underlying the findings in their manuscript fully available?

Reviewer #1: Yes

4. Is the manuscript presented in an intelligible fashion and written in standard English?

Reviewer #1: Yes

5. Review Comments to the Author

Reviewer #1: This is an interesting and important paper that uses modeling to explore Bd dynamics in the Atelopus cruciger system. The authors make important contributions to the field. They find evidence that the population of free Bd zoospores in environmental reservoirs are more likely to push populations from endemic coexistence to extinction than changes in the vulnerability of adults to chytridiomycosis. Their models are informed by a wealth of field-collected demographic and ecological data in this system. I found their study to be well written and presented. My only major critique is that I believe a substantial number of important and relevant studies in this system are not discussed or cited in this paper. Their omission is notable given the immediate relevance of these papers and their prominence in this field. I think the authors should more thoroughly consider how their findings are situated in relation to previous modeling efforts of Bd, especially when it comes to modeling zoospore pools, lethal infection intensities, and comparing how changes in these affect extinction dynamics. Below I point out some minor errors in the paper and list some important citations I believe were erroneously omitted.

Line 3 – This figure “500 amphibian species, including 90 presumed extinctions [1].” is not actually supported by sound evidence. See Lambert et al. 2020 response to the citation listed (DOI: 10.1126/science.aay1838). I recommend modifying this citation to omit these specific, inflated numbers or also cite the response as “[1], but see [Lambert et al. 2020]”.

Line 17 – The citation for A. palmatus should be [13] not [12].

Line 25 – Should be consistent in either using the abbreviated Bd or the longer B. dendrobatidis throughout the paper.

Figure 3 – It would be helpful to include in the figure legend what the colors correspond to.

Line 183 – The opening statement of the discussion should have citations for the facts it is listing.

Line 195 – Typo “estimated in” should be “estimated to be”

Line 218 – The sentence “Harlequin toads tend to be short-lived.” needs some qualifiers. For example, harlequin toads in captivity can live up to 10 years. The citations used imply that the authors mean to say that “Harlequin toads coexisting with Bd in the wild tend to be short-lived.”

Line 222 – The sentence “In the presence of a highly pathogenic Bd, only uninfected toads are likely to reproduce.” Is there data to support this assertion? If so, you should cite here and perhaps explain in a bit more detail.

Line 224 – There is a typo here. “when adults” should be “when individuals are adults.”

Line 238 – The citation here is a review and should be cited as such (i.e., [reviewed in 36]).

Some seminal papers exploring Bd dynamics are not discussed or cited here are would add important context to this study. For example Wilbur et al. 2017 (https://doi.org/10.1111/ele.12814) used a modelling approach to find that including an environmental zoospore pool increased R0 for R. muscosa‐Bd systems. They also found that Bd‐induced extinction risk was more sensitive to host resistance and tolerance than to the transmission dynamics of Bd. The similarities/differences between this study and that should be explored in the discussion. Additionally, the paper Vredenburg et al. 2010 (https://doi.org/10.1073/pnas.0914111107) is an important citation for estimating the lethal dose of Bd on an amphibian and should at least be mentioned or cited within given the estimation of lethal dose in this paper. Further, the findings of Briggs et al. 2010 (https://doi.org/10.1073/pnas.0912886107

) should be discussed in this paper. I think these papers should be included in the introduction and further explored in the discussion.

6. PLOS authors have the option to publish the peer review history of their article (what does this mean?). If published, this will include your full peer review and any attached files.

Reviewer #1: No

---

## [Author Response · Author response to Decision Letter 0]

6 May 2021

Response to Reviewers:

Line 3 – This figure “500 amphibian species, including 90 presumed extinctions [1].” is not actually supported by sound evidence. See Lambert et al. 2020 response to the citation listed (DOI: 10.1126/science.aay1838). I recommend modifying this citation to omit these specific, inflated numbers or also cite the response as “[1], but see [Lambert et al. 2020]”.

We omitted figures and changed the first sentence to the more general statement: “Chytridiomycosis………has been linked to the local extirpation and extinction of amphibian populations worldwide, although the exact number of species affected remains controversial (Scheele et al 2019; Lambert et al.2020)”. We directed the reader to the suggested references to enquire further about the estimated number of species affected.

Line 17 – The citation for A. palmatus should be [13] not [12]. 

Done

Line 25 – Should be consistent in either using the abbreviated Bd or the longer B. dendrobatidis throughout the paper.

We used Bd throughout the paper.

Figure 3 – It would be helpful to include in the figure legend what the colors correspond to.

I am not sure what the reviewer means. The figure legend reads: “Blue pixels represent combinations of parameters for which coexistence is the most likely outcome, green denotes pairs for which Bd is most likely to fail in establishing and red denotes pairs for which Bd is more likely to extirpate the toad population. Please let me know if you think we need to add more information.

Line 183 – The opening statement of the discussion should have citations for the facts it is listing.

We added seven references to the opening statement:

“The high impact of chytridiomycosis at high elevations (Catenazzi et al. 2014; Kriger & Hero 2008; Scheele et al. 2019; Woodhams & Alford 2005) and the recent discovery of recovering populations in lowlands (Grogan et al. 2016; Phillott et al. 2013; Rodríguez-Contreras et al. 2008) suggest that warm temperatures may promote the coexistence of some amphibian species with the pathogenic fungus Bd (Puschendorf et al. 2011; Zumbado-Ulate et al. 2014).

Line 195 – Typo “estimated in” should be “estimated to be”. 

Done

Line 218 – The sentence “Harlequin toads tend to be short-lived.” needs some qualifiers. For example, harlequin toads in captivity can live up to 10 years. The citations used imply that the authors mean to say that “Harlequin toads coexisting with Bd in the wild tend to be short-lived.” 

We modified this sentence as follows: “Harlequin toads tend to be short-lived in the wild” Although in captivity harlequin toads can live in up to 10 years, mark-recapture studies suggest that in the wild uninfected adults of A. cruciger and A. zeteki have average life expectancy of less than one year. Thus, even in the absence of Bd, harlequin toads tend to be short-lived.

In general amphibians in captivity can last many years more than in the wild. I think that there are records of R. marina of about 40 years, but I doubt that they can live that long in the wild. 

Line 222 – The sentence “In the presence of a highly pathogenic Bd, only uninfected toads are likely to reproduce.” Is there data to support this assertion? If so, you should cite here and perhaps explain in a bit more detail.

There is no data demonstrating that infected toads cannot reproduce. However, given that they die within weeks after infection and amplexus in these toads can last months it seems unlikely that they can get infected and reproduce. I elaborated on this idea in the new version. 

Line 224 – There is a typo here. “when adults” should be “when individuals are adults.” 

Done.

Line 238 – The citation here is a review and should be cited as such (i.e., [reviewed in 36]). 

Done

Some seminal papers exploring Bd dynamics are not discussed or cited here and would add important context to this study. 

For example Wilbur et al. 2017 (https://doi.org/10.1111/ele.12814) used a modelling approach to find that including an environmental zoospore pool increased R0 for R. muscosa‐Bd systems. They also found that Bd‐induced extinction risk was more sensitive to host resistance and tolerance than to the transmission dynamics of Bd. The similarities/differences between this study and that should be explored in the discussion. 

Thanks for pointing out Wilbert et al.’s paper; it was enlightening. We mentioned their findings on the importance of resistance and tolerance on risk extinction in the Introduction section (lines X) Also, we included a whole paragraph explaining differences between these two models as I believe the comparison is highlight the need for context-specific modeling. 

Additionally, the paper Vredenburg et al. 2010 (https://doi.org/10.1073/pnas.0914111107) is an important citation for estimating the lethal dose of Bd on an amphibian and should at least be mentioned or cited within given the estimation of lethal dose in this paper. 

Vredenburg et al., 2010 was added to the list of sources of lethal doses used on Table 1 in the Appendix.

Further, the findings of Briggs et al. 2010 (https://doi.org/10.1073/pnas.0912886107

) should be discussed in this paper. I think these papers should be included in the introduction and further explored in the discussion.

Briggs et al.’s paper was added to the list of papers supporting the importance of biotic or abiotic reservoirs in the extinction risk of host populations in the introduction and discussion sections.

---

## [Decision Letter · Decision Letter 1]

23 Jun 2021

PONE-D-21-04928R1

Living with the pathogenic chytrid fungus: exploring mechanisms of coexistence in the harlequin toad Atelopus cruciger.

PLOS ONE

Dear Dr. Lampo,

Thank you for submitting your manuscript to PLOS ONE. After careful consideration, we feel that it has merit but does not fully meet PLOS ONE’s publication criteria as it currently stands. Therefore, we invite you to submit a revised version of the manuscript that addresses the points raised during the review process.

The authors have done a proper job in revising their paper. It satisfied the referee and myself. So, the paper can be accepted, but there are a few minor thigs that need some attention why this is a technical "minor revision".

We look forward to receiving your revised manuscript.

Kind regards,

Stefan Lötters

Academic Editor

PLOS ONE

Journal Requirements:

Reviewers' comments:

Reviewer's Responses to Questions

**Comments to the Author**

1. If the authors have adequately addressed your comments raised in a previous round of review and you feel that this manuscript is now acceptable for publication, you may indicate that here to bypass the “Comments to the Author” section, enter your conflict of interest statement in the “Confidential to Editor” section, and submit your "Accept" recommendation.

Reviewer #1: All comments have been addressed

2. Is the manuscript technically sound, and do the data support the conclusions?

Reviewer #1: Yes

3. Has the statistical analysis been performed appropriately and rigorously? 

Reviewer #1: Yes

4. Have the authors made all data underlying the findings in their manuscript fully available?

Reviewer #1: Yes

5. Is the manuscript presented in an intelligible fashion and written in standard English?

Reviewer #1: Yes

6. Review Comments to the Author

Reviewer #1: I am satisfied with the changes made by the authors and feel that all comments have been adequately addressed. I especially like the new discussion paragraph discussing how the model describing this system compares to that built to describe the mountain yellow-legged frog system. I did notice a few typos/grammatical errors (mostly in the new text additions) that I will note below. Nice work!

In the abstract:

- replace "a disease produced by" with "a disease caused by"

- "most of the others are in high risk of extinction" should be "most of the others are at high risk of extinction"

Line 267: "have proved to be" should be "has proved to be"

Line 271 (and throughout this paragraph): The common name for Rana muscosa is "Mountain Yellow-legged Frog" not "yellow-legged frog"

Line 276: The sentence that reads: "That is, not only recruiting health juveniles contribute to the susceptible population but also

infected adults that have cleared infection." should be "That is, not only does the recruitment of healthy juveniles contribute to the susceptible population, but also the presence of infected adults that have cleared infection."

7. PLOS authors have the option to publish the peer review history of their article (what does this mean?). If published, this will include your full peer review and any attached files.

Reviewer #1: No

---

## [Author Response · Author response to Decision Letter 1]

23 Jun 2021

In the abstract:

- replace "a disease produced by" with "a disease caused by"

Replaced as suggested.

- "most of the others are in high risk of extinction" should be "most of the others are at high risk of extinction"

Replaced as suggested.

Line 267: "have proved to be" should be "has proved to be"

Replaced as suggested.

Line 271 (and throughout this paragraph): The common name for Rana muscosa is "Mountain Yellow-legged Frog" not "yellow-legged frog"

Replaced as suggested.

Line 276: The sentence that reads: "That is, not only recruiting health juveniles contribute to the susceptible population but also infected adults that have cleared infection." should be "That is, not only does the recruitment of healthy juveniles contribute to the susceptible population, but also the presence of infected adults that have cleared infection."

Replaced as suggested.

---

## [Editor Report · Decision Letter 2]

28 Jun 2021

Living with the pathogenic chytrid fungus: exploring mechanisms of coexistence in the harlequin toad Atelopus cruciger.

PONE-D-21-04928R2

Dear Dr. Lampo,

We’re pleased to inform you that your manuscript has been judged scientifically suitable for publication and will be formally accepted for publication once it meets all outstanding technical requirements.

Kind regards, listo el pollo!

Stefan Lötters

Academic Editor

PLOS ONE
---

## [Editor Report · Acceptance letter]

5 Jul 2021

PONE-D-21-04928R2 

Living with the pathogenic chytrid fungus: exploring mechanisms of coexistence in the harlequin toad *Atelopus cruciger*. 

Dear Dr. Lampo:

I'm pleased to inform you that your manuscript has been deemed suitable for publication in PLOS ONE. Congratulations! Your manuscript is now with our production department. 

Kind regards, 

on behalf of

Prof. Dr. Stefan Lötters 

Academic Editor

PLOS ONE